# Seroprevalence of Anti-SARS-CoV-2 Antibodies in Cats during Five Waves of COVID-19 Epidemic in Thailand and Correlation with Human Outbreaks

**DOI:** 10.3390/ani14050761

**Published:** 2024-02-29

**Authors:** Suporn Thongyuan, Jeeraphong Thanongsaksrikul, Potjanee Srimanote, Wallaya Phongphaew, Piyaporn Eiamcharoen, Naris Thengchaisri, Angela Bosco-Lauth, Nicola Decaro, Rungrueang Yodsheewan

**Affiliations:** 1Department of Veterinary Public Health, Faculty of Veterinary Medicine, Kasetsart University, Kamphaeng Saen Campus, Nakhon Pathom 73140, Thailand; fvetspty@ku.ac.th; 2Graduate Program in Biomedical Sciences, Faculty of Allied Health Sciences, Thammasat University, Pathumtani 12121, Thailand; jeeraphong.t@allied.tu.ac.th (J.T.); psrimanote01@yahoo.com.au (P.S.); 3Department of Pathology, Faculty of Veterinary Medicine, Kasetsart University, Bangkok 10900, Thailand; wallaya.p@ku.th (W.P.); piyaporn.e@ku.th (P.E.); 4Department of Companion Animal Clinical Sciences, Faculty of Veterinary Medicine, Kasetsart University, Bangkok 10900, Thailand; ajnaris@yahoo.com; 5Department of Biomedical Sciences, Colorado State University, 3107 W Rampart Road, Fort Collins, CO 80523, USA; angela.bosco-lauth@colostate.edu; 6Department of Veterinary Medicine, University of Bari, Strada Provinciale per Casamassima, Valenzano, 70010 Bari, Italy; nicola.decaro@uniba.it

**Keywords:** COVID-19, SARS-CoV-2, indirect ELISA, surrogate viral neutralization assay, seroprevalence, correlation, Thai, cat

## Abstract

**Simple Summary:**

This study investigated the transmission of COVID-19 from humans to animals, specifically highlighting a case where a veterinarian contracted the virus from an infected cat. The objectives included examining the presence of SARS-CoV-2 antibodies in Thai cats during various episodes of the outbreak, evaluating the effectiveness of a modified human commercial test kit for screening SARS-CoV-2 antibodies in cats, and investigating the correlation between cat infections and human epidemic episodes. Utilizing 1107 cat serum samples, the results demonstrated a seropositive rate of 22.67%, aligning with trends observed in humans. The cPass™ neutralization test revealed a validated 3.99% seropositivity rate. While distinct patterns were observed among epidemic waves, overall variation was revealed across provinces. Particularly, Samut Sakhon demonstrated a robust positive correlation between the proportion of positive cat sera and human prevalence. This study underscores the importance of continuous surveillance to comprehend the dynamics of SARS-CoV-2 transmission in human and feline populations.

**Abstract:**

Human-to-animal SARS-CoV-2 transmission was observed, including a veterinarian contracting COVID-19 through close contact with an infected cat, suggesting an atypical zoonotic transmission. This study investigated the prevalence of SARS-CoV-2 antibodies in cats during human outbreaks and elucidated the correlation between cat infections and human epidemics. A total of 1107 cat serum samples were collected and screened for SARS-CoV-2 antibodies using a modified indirect ELISA human SARS-CoV-2 antibody detection kit. The samples were confirmed using a cPass™ neutralization test. The SARS-CoV-2 seropositivity rate was 22.67% (199/878), mirroring the trend observed in concomitant human case numbers. The waves of the epidemic and the provinces did not significantly impact ELISA-positive cats. Notably, Chon Buri exhibited a strong positive correlation (r = 0.99, *p* = 0.009) between positive cat sera and reported human case numbers. Additionally, the cPass™ neutralization test revealed a 3.99% (35/878) seropositivity rate. There were significant differences in numbers and proportions of positive cat sera between epidemic waves. In Samut Sakhon, a positive correlation (r = 1, *p* = 0.042) was noted between the proportion of positive cat sera and human prevalence. The findings emphasize the need for ongoing surveillance to comprehend SARS-CoV-2 dynamics in both human and feline populations.

## 1. Introduction

From December 2020 to March 2022, Thailand experienced five waves of COVID-19 outbreaks, with the initial occurrence in March to May 2020. The subsequent waves transpired from December 2020 to February 2021, April to June 2021, July to December 2021, and January to March 2022, as documented by Puenpa et al. (2022) [1]. Initially concentrated in Bangkok during the early stage, COVID-19 nationwide dissemination ensued primarily due to a significant migration following the March 2020 Bangkok shutdown [2]. Examining instances of zoonosis, a veterinarian contracted COVID-19 through close contact with an infected cat who had resided with COVID-19 human patients, highlighting the atypical nature of this transmission mode [3]. Empirical evidence suggests heightened susceptibility of cats to SARS-CoV-2, with laboratory experiments showing they can transmit the virus [4]. Laboratory experiments have further confirmed that cats can transmit the SARS-CoV-2 virus to their feline counterparts [5,6]. Consequently, it is plausible for cats residing in households with COVID-19 patients to become integral components of the transmission pathway of the COVID-19 virus, constituting a human-to-animal-to-human transmission cycle.

Surveillance of COVID-19 infections in the feline population is crucial for future epidemic control and prevention planning. There is a moderate likelihood of cats contracting COVID-19 in close contact conditions with their infected owners, as per expert opinion [7]. The necessity to segregate cats from COVID-19 patients becomes a pivotal measure to avert and mitigate transmission from humans to felines [6,8,9]. Continuous scientific inquiry and systematic data collection on COVID-19 infections in cats, especially during human COVID-19 outbreaks, facilitate early detection in the feline demographic and provide essential insights for readiness in preventing and monitoring the spread of COVID-19 among cats [10,11,12]. Notably, during periods of endemic COVID-19 when local testing measures or disease reporting may decrease [11,12], utilizing the cat population as sentinels for disease surveillance emerges as a viable alternative for monitoring COVID-19 within both human and animal populations [13,14], particularly among companion animals. In this circumstance, serological assay seems to be the most appropriate method for large-scale screening tests and surveillance [15]. Moreover, the serological assay can diagnose recent SARS-CoV-2 infection, which facilitates determining the actual disease burden [16]. Recently, Udom et al. (2021) reported the utility of a human anti-N IgG ELISA commercial kit to detect the anti-N antibodies and surrogate virus to detect anti-S neutralizing antibodies in dogs and cats during the Thailand epidemic from April to December 2020 [17]. Moreover, the seroprevalence against SARS-CoV-2 was investigated. It was found that 1.66% and 0.36% of dogs and cats, respectively, were positive for SARS-CoV-2 antibodies. However, none of them were found positive for anti-S neutralizing antibodies by the cPass™ neutralization test. After this period, Thailand was attacked by four waves of COVID-19 outbreaks [1] predominated by different SARS-CoV-2 clades/variants. Regarding virus adaptation and natural selection, the different SARS-CoV-2 clades/variants with distinct characteristics made it worth re-assessing the utility of serological tests for diagnosing SARS-CoV-2 infection in domestic animals during the subsequent waves of Thailand’s COVID-19 epidemics.

There was a concern about the discrepancy results between the screening test, anti-N antibody ELISA, and the confirmatory test, anti-S neutralizing antibodies [17]. Moreover, cross-reactivity between antibodies to N proteins of feline coronavirus and SARS-CoV-2 was reported [18]. This may be explained by the conserved N sequence among coronaviruses [18]. Therefore, anti-S1 and anti-S1 RBD antibody assays were recommended for screening SARS-CoV-2 antibodies in cats [18]. However, it has been known that newly emerged SARS-CoV-2 variants are characterized by sets of mutations particularly in a coding sequence of S protein [19]. The mutated S protein can impact virus immunogenicity, which affects the host’s humoral immune responses. This issue may compromise the diagnostic accuracy of anti-S antibody assay for screening cat sera. Therefore, the ultimate objective of this study was to re-assess the utility of the modified human test kit, namely, anti-S1 RBD IgG ELISA for diagnosing SARS-CoV-2 infection in cats during five waves of Thailand’s COVID-19 epidemics. Additionally, seroprevalence and the correlation between SARS-CoV-2 infections in cats and human COVID-19 epidemics in Thailand were investigated. The significance of this research was to gain insight into the feasibility of employing cats as candidate sentinels for future disease surveillance during COVID-19 human outbreaks.

## 2. Materials and Methods

A total of 1107 cat serum samples were collected between December 2020 and March 2022 in five major provinces according to the high reported numbers of confirmed human cases: Bangkok, Pathum Thani, Chon Buri, Samut Sakhon, and Phuket. Two hundred and ninety-two cat serum samples collected from Nakhon Pathom and Bueng Kan provinces (non-outbreak areas) before the report of COVID-19 outbreaks were also included in this study. The studied areas are shown in the map of Thailand (Figure 1). The numbers of cat sera collected during the 1st to 5th waves of the infection and details on the reported human cases are shown in Table 1. Three positive control samples were derived from cats diagnosed with COVID-19, as confirmed by PCR and the plaque reduction neutralization test. These positive samples originated from Italy (two samples) and the United States (one sample) and were kindly provided by Dr. Nicola Decaro and Dr. Angela Bosco-Lauth. Commercial pooled normal cat sera procured from 200 cats collected prior to the COVID-19 outbreak (Nordic-MUBio, Inc., Susteren, The Netherlands) were used as the negative control.

Human SARS-CoV-2 S1 RBD IgG antibody ELISA kit (CUSABIO, Wuhan, Hubei, China) was used to perform the indirect ELISA screening test according to the manufacturer’s instructions, except the secondary antibody was substituted with horseradish peroxidase-conjugated goat anti-feline IgG (H + L) (Invitrogen, Camarillo, CA, USA). For optimization, the three confirmed SARS-CoV-2 infection-positive sera and the commercial pooled normal cat sera (negative control) were diluted 1:100, 1:200, 1:400, and 1:800 in the sample diluent. One hundred microliters of the diluted samples were added to each well of commercial SARS-CoV-2 S1 RBD coated assay plates and incubated at 37 °C for 30 min. Wells were washed three times with 1× wash buffer, and 1:10,000 HRP-conjugated goat anti-feline IgG (H + L) was added. After 30 min of incubation at 37 °C, the assay plate was then washed five times, followed by adding TMB substrate and incubation at 37 °C for 20 min. The absorbance was measured at wavelengths of 450 and 570 nm using Synergy LX Multi-Mode Microplate Reader (Agilent Biotek, Santa Clara, CA, USA) after adding a stop solution. The 1:400 dilution was best to differentiate SARS-CoV-2-positive and -negative cat sera. To evaluate potentially cross-reacting cat antibodies, a total of 136 cat serum samples from 18 categories and 156 cat serum samples in non-outbreak areas were tested (Appendix A). All sera were diluted at 1:400, and modified SARS-CoV-2 S1 RBD ELISA was performed as described previously.

Screening of SARS-CoV-2 antibody in cat sera using modified SARS-CoV-2 S1 RBD ELISA was performed. In brief, the SARS-CoV-2 positive control, negative control sera (commercial pooled normal cat sera), and tested serum samples were diluted 1:400 in the sample diluent. Modified ELISA was performed as described previously. The cut-off value (OD_sample_/OD_negative_) was determined by ROC curve analysis and computed by Youden’s index [20,21], as shown in preliminary data (Appendix A). Samples with a value of OD_sample_/OD_negative_ equal to or greater than 2.02 were considered positive for modified SARS-CoV-2 S1 RBD ELISA.

Cat sera were confirmed for the presence of SARS-CoV-2 antibody by the cPass™ SARS-CoV-2 Neutralization Antibody Detection Kit (Nanjing GenScript Diagnostics Technology, Nanjing, Jiangsu, China) following the manufacturer’s instructions [22]. In brief, serum samples and positive and negative controls were diluted 1:10 in the sample dilution buffer. Subsequently, 60 µL of the diluted samples was incubated with an equal volume of 1:3000 HRP conjugated RBD at 37 °C for 30 min. Then, 100 µL of the mixture was transferred to the ACE2-capture plate and incubated for 15 min at 37 °C. Following four washes with 1× wash buffer, TMB substrate (100 µL/well) was added, and the plate was then incubated at room temperature for 15 min in a dark, humidified box. A stop solution (50 µL/well) was finally added, and the OD was measured at a wavelength of 450 nm using Synergy LX Multi-Mode Microplate Reader. The percentage of inhibition (% inhibition) was calculated according to the manufacturing instructions. Inhibition equal to or greater than 20% was considered positive for SARS-CoV-2 neutralizing antibodies.

Statistical analyses were performed using R Statistical Software (version 4.3.1; R Foundation for Statistical Computing, Vienna, Austria). Descriptive statistics were used to determine the prevalence of seropositivity in cats. ANOVA and Kruskal–Wallis H tests were performed to compare the differences between the number of positive cat serum samples and the percentage of seropositive cats by different waves of the epidemic and provinces. The effect of epidemic waves and provinces on the number of positive cat serum samples and the proportion of seropositive cats was analyzed by two-way ANOVA. Pearson’s and Spearman correlation tests were performed to assess the relationship between the prevalence of human cases (reported cases per 100,000 population) and the prevalence of seropositivity in cats (confirmed by cPass™ neutralization test). Statistical significance was assessed at *p* < 0.05.

## 3. Results

Cat serum samples collected from the included provinces were screened for antibodies against SARS-CoV-2 by the modified SARS-CoV-2 S1 RBD ELISA method, and confirmed by the cPass™ neutralization test. The distribution of seropositive cat sera in relation to the epidemic waves and provinces is shown in Table 1 and Figure 1. The number of reported human cases across the five provinces during the five outbreak waves in Thailand is shown in Figure 2 [23]. Among these five provinces, the number of reported human cases tended to increase over time; the highest number of reported cases was shown during the fourth wave of the epidemic. The highest number of human cases was reported in the Bangkok–Pathum Thani area, followed by Chon Buri, Samut Sakhon, and Phuket.

### 3.1. Prevalence of SARS-CoV-2 Antibody in Cats by Modified SARS-CoV-2 S1 RBD ELISA

Our modified SARS-CoV-2 S1 RBD ELISA had high accuracy (AUC = 0.903; *p* < 0.0001) and good effectiveness (Youden’s index = 0.788) for the detection of SARS-CoV-2 antibodies in cat sera (Appendix A). Of the 1107 feline serum samples, 878 samples were from areas affected by the human COVID-19 outbreaks, and 22.67% (199/878) of those demonstrated positivity against SARS-CoV-2 by indirect ELISA during the second to fifth epidemic waves in Thailand. Notably, the trends in the number of positive feline serum samples and the proportion of seropositive cats mirrored those observed in human cases, as illustrated in Figure 3. Surprisingly, 229 cat serum samples from non-outbreak areas exhibited a seropositivity rate of 17% (39/229) (Table 1).

In the specific context of the Bangkok–Pathum Thani area, during the second wave of the epidemic with reported human cases at 2943 (44.38 cases per 100,000 population), the percentage of ELISA-positive cats was 12.5%, then escalating to 15.22% in the third wave and reaching 24% in the fourth wave. Only one serum sample was collected during the fifth wave, yielding a negative result for ELISA. In Chon Buri, no ELISA-positive case was detected during the second and third waves, aligning with reported human cases of 662 (42.61 cases per 100,000 population) and 11,254 (724.40 cases per 100,000 population), respectively. However, a substantial proportion of seropositive cats (39.58%) emerged in the fourth wave, coinciding with a surge in reported human cases to 102,974 (6616.67 casesper 100,000 population), followed by a decline to 11.64% in the fifth wave. In Samut Sakhon, the proportion of seropositive cats was 12.50% in the third wave, rising twofold to 25.94% in the fourth wave, concomitant with a surge in reported human cases to 82,921 (15,025.54 cases per 100,000 population). The highest percentage of seropositivity, reaching 30.65%, was noted in the fifth wave, despite a 50% reduction in reported human cases. Notably, in Phuket, no seropositive cases were detected during the second wave. However, the highest proportion of seropositive cats (43.8%) was observed in the third wave, correlating with 769 reported human cases (189.77 cases per 100,000 population), followed by proportions of 17.02% and 24.41% in the fourth and fifth waves, respectively. The waves of the epidemic and provinces did not have a statistically significant effect on the proportion of ELISA-positive cats (*p* = 0.329 and *p* = 0.802, respectively). The comparison of the numbers of ELISA-positive cats by the epidemic waves is shown in Figure 3A. There was no statistically significant difference between the numbers of ELISA-positive cats for SARS-CoV-2 by different waves of the epidemic (H (3) = 7.723, *p* = 0.052). Similarly, the means of proportions of seropositive cats for SARS-CoV-2 were not statistically significantly different among the different waves of the epidemic (F (3) = 1.632, *p* = 0.238) (Figure 3B). Regarding the comparison among provinces, there was no statistically significant difference between the number of seropositive cats for SARS-CoV-2 in different provinces (H (3) = 0.897, *p* = 0.826) (Figure 3C). Similarly, no statistically significant difference between the proportion of seropositive cats for SARS-CoV-2 in different provinces (H (3) = 0.471, *p* = 0.709) was found (Figure 3D).

A significant positive correlation was observed between the number of positive cat serum samples and the prevalence of reported human cases in Chon Buri province (r = 0.99, *p* = 0.009) (Figure 4B). Simple linear regression was used to test if the prevalence of reported human cases would significantly predict the number of ELISA-positive cat serum samples. The fitted regression model was y = 0.004x − 12.42. The overall regression was statistically significant (R^2^ = 0.93, F (1, 1) = 27.43, *p* < 0.01. It was found that the prevalence of reported human cases significantly predicted the number of ELISA-positive cat serum samples (β = 0.004, *p* < 0.01). Conversely, in Bangkok–Pathum Thani, Samut Sakhon, and Phuket, strong correlations were observed between the numbers of positive cat sera and prevalence of reported human cases during the considered period (r = 0.77, 0.98, and 0.89, respectively); however, these correlations were not statistically significant (*p* > 0.05) (Figure 4A,C,D).

### 3.2. Confirmation of SARS-CoV-2 Infection in Cats Using cPass^TM^ Neutralization Test

In total, the percentage of cat sera collected from five provinces that were true seropositive cases of SARS-CoV-2 according to the cPass™ test was 3.99% (35/878). The trends of the amount of positive cat sera and the proportion of positive cats were the same as in human cases (Figure 2). Among the 39 seropositive cat samples detected by indirect ELISA out of 229 serum samples collected from non-outbreak areas, only one cat serum sample from Nakhon Pathom province (0.44%) tested positive with the cPass™ test. 

In the Bangkok–Pathum Thani area, 213 cat serum samples were collected throughout the five waves of the epidemic. During the second wave, when the number of reported human cases was 2943 (44.38 per 100,000 population), none of the cat serum samples were found seropositive by the cPass™ test. However, during the third wave of the epidemic, when the number of reported human cases was 125,788 (1896.80 cases per 100,000 population), the percentage of true positive cat sera was 2.17%, and then increased to 4.0% during the fourth wave. Only one cat serum sample collected in the fifth wave showed negative results using both indirect ELISA and cPass™ (Table 1) tests. In Chon Buri, with 203 cat serum samples collected over the five waves, no true seropositive cases were detected during the second and third waves (662 and 11,254 reported human cases and 42.61 and 724.40 cases per 100,000 population, respectively). However, a notable surge in true seropositive samples (2.08%) occurred during the fourth wave, aligning with a rise in reported human cases to 102,974 (6616.67 cases per 100,000 population), followed by an increase in true seropositive samples to 3.42% in the fifth wave. In Samut Sakhon, where in total 262 cat serum samples were collected, no true seropositive samples were found during the third wave (12,407 reported human cases, 2248.19 per 100,000 population), but the percentage of true seropositive samples increased to 7.08% (15/212) in the fourth wave (82,921 reported human cases, 15,025.54 cases per 100,000 population) and subsequently decreased to 2.38% (1/42) in the fifth wave, coinciding with an approximately 50% reduction in reported human cases. Notably, the trend presented by the confirmation test was different from that of the screening test. Finally, in Phuket, with 200 cat serum samples collected, no true seropositive samples were detected during the second and third waves. However, the highest proportion of true seropositives occurred in the fourth wave (8.51%) with 18,164 reported human cases (4482.34 cases per 100,000 population), followed by a decrease to 1.57% in the fifth wave. Similarly, the trend presented by the confirmation test was different from that of the screening test. 

A comparison of the number of true seropositive samples throughout the waves of the epidemic is shown in Figure 5A. There was a statistically significant difference between the number of samples found positive for SARS-CoV-2 (true seropositive) using cPass™ during different waves of the epidemic (H (3) = 9.268, *p* = 0.026). The median numbers of positive viral neutralization cat sera during the second to fifth waves were 0, 0, 5, and 1.5, respectively. The results show that the median number of positive cat serum samples in the fourth wave was significantly higher than in the third wave. Similarly, the median percentage of cat serum samples found positive for SARS-CoV-2 using cPass™ was statistically significantly different during the different waves of the epidemic (F (3) = 9.194, *p* = 0.027) (Figure 5B). The median percentages of positive viral neutralization cat sera during the second to fifth waves were 0, 0, 5.54, and 1.98, respectively. The results show that the median number of true seropositive cat serum samples in the fourth wave was significantly higher than in the second and third waves. Regarding the comparison among the provinces, there was no statistically significant difference between the number of true seropositive serum samples for SARS-CoV-2 in different provinces (H (3) = 0.383, *p* = 0.944) (Figure 5C). Similarly, no statistically significant difference was found between the percentages of SARS-CoV-2 seropositive samples detected by cPass™ in different provinces (H (3) = 0.616, *p* = 0.893) (Figure 5D). 

A significant positive correlation was observed between the percentage of true positive cat sera and the prevalence of reported human cases in Samut Sakhon province (r = 1, *p* = 0.042) (Figure 6G); the correlation between the number of true positive cat sera and the prevalence of reported human cases was not significantly different in all areas (Figure 6A–D). Simple linear regression was used to test if the prevalence of reported human cases significantly predicted the proportion of true positive cat sera. The fitted regression model was y = 0.0006x − 1.41. The overall regression was statistically significant (R^2^ = 0.99, F (1, 1) = 234.1, *p* = 0.04). It was found that the prevalence of reported human cases significantly predicted the proportion of true positive cat sera (β = 0.0006, *p* = 0.04). Additionally, in Bangkok–Pathum Thani, Chonburi, and Phuket, there was a strong correlation (r = 0.70, r = 0.87, and ρ = 0.74, respectively) between the percentages of true positive cat sera and the prevalence of reported human cases during the considered period; however, it was not statistically significant (*p* > 0.05) (Figure 6E,F,H). 

Overall, the implementation of the SARS-CoV-2 ELISA screening test with a cut-off value of 2.02 gave 100% sensitivity and 78.86% to 89.18% specificity when confirmed by the cPass™ neutralization test, which was almost comparable to the estimated values (96% sensitivity and 89.89% specificity).

## 4. Discussion

SARS-CoV-2 infection in companion animals (dogs and cats) appears to be associated with the reported COVID-19-positive status of the humans in their shared households [24]. In Germany, the seroprevalence in cats was doubled in accordance with the rise in reported human cases, indicating a continuous occurrence of transspecies transmission from infected owners to their cats [25]. In this study, the human SARS-CoV-2 S1 RBD IgG antibody ELISA kit was modified and employed to screen feline sera, followed by the confirmatory test using neutralization antibody detection (cPass™). The investigation of the prevalence of SARS-CoV-2 antibodies in Thai cats during the observed COVID-19 epidemic waves yielded similar notable findings with implications for public health and veterinary considerations. The study revealed a 22.67% seropositivity rate in cat serum samples, determined through indirect ELISA. This percentage closely mirrors the trends observed in human cases, reinforcing the interconnectedness of the viral dynamics between the two populations.

Utilizing modified human screening COVID-19 serological tests (anti-S1 RBD IgG ELISA) for SARS-CoV-2 infection surveillance in the cat population presents potential benefits, although the confirmed seropositivity rate via cPass™ declined to 3.99% while that of the screening test (ELISA) was 22.67%. This reduction was attributed to a substantial number of false positive cases identified through the anti-S1 RBD indirect ELISA test. Similarly, a discrepancy between the results of the screening test using anti-N IgG ELISA and cPass™ was reported by Udom et al. (2021) [17]. The study of Zhao et al. (2021) showed that feline coronavirus could serologically cross-react with the N protein of SARS-CoV-2 [18]. In our study using the S1 RBD IgG antibody ELISA, interference from non-specific cross-reactivity of cat sera with other viral infections, including feline leukemia virus and feline panleukopenia virus, as indicated in preliminary data (Appendix A), might influence the results; however, this observation could not be definitively confirmed. Additionally, misdiagnoses from the screening test occurred under unknown circumstances, with cat sera associated with chronic kidney disease and those sampled in non-outbreak areas contributing to false positive results. The reliance on ELISA alone may prove inadequate for assessing relationships during disease outbreaks, as revealed in the results, where differences in analyzing the two datasets with ELISA were predominantly non-significant, except for a significant finding at the median of the numbers of ELISA positive sera of Chon Buri, attributed to a high false positive rate. In contrast, cPass™ demonstrated significance for the percentages of the positive sera in Samut Sakhon, offering more reliable and true positive results. This underscores the importance of employing multiple testing methods for a comprehensive and accurate assessment, particularly in the context of disease outbreaks.

The single true positive cat serum samples collected in non-outbreak areas, specifically Nakhon Pathom province, could suggest an underrepresented transmission between humans and cats and could potentially indicate positive human cases in such areas before they are officially declared as an outbreak area. However, this assumption prompted further investigation. After the fourth wave, the number of positive cat serum samples determined by both tests decreased or did not show a similar trend as in human epidemics. As the virus strain in humans was transitioning to Omicron during this phase [26], we speculated that infection with the Omicron strain after the fourth wave could have been less contagious to cats as the virus evolved [27].

The study underscores the importance of continuous surveillance for SARS-CoV-2 in both human and feline populations, revealing correlations in specific provinces and highlighting cats as potential sentinels for human infections. While owning a domestic cat appears to pose a low to insignificant risk for additional human infections [3] based on cat-to-human interactions, separating cats from infected individuals can aid in preventing further transmission [28,29]. The decline in cat cases after the fourth wave, amid rising human cases, suggests a potential association with the emergence of the Omicron variant. This temporal alignment with Omicron’s introduction in Thailand implies viral evolution favoring increased human adaptability [30,31,32] while reducing feline susceptibility [33,34]. Molecular and epidemiological investigations may shed light on the intricate interplay between viral evolution and host susceptibility [35,36,37]. Although cPass™-positive cat proportions remained consistent across epidemic waves, the robust correlation in Samut Sakhon underscores the significance of considering regional variations, providing valuable insights into localized factors influencing transmission dynamics between humans and cats across five major provinces.

Despite these significant findings, the study acknowledges certain limitations, including the need for further research to establish causation and the potential influence of confounding variables. Furthermore, the proportion of the positive cat population sampled in this study might not precisely reflect the dynamics seen in human epidemic waves because lockdown measures prevented sample collection at the peak of the epidemic wave [2,38]. Additionally, the lack of statistical significance in some correlations, notably in Bangkok–Pathum Thani and Phuket, highlights the complexity of the relationship between human and feline infections, necessitating continued investigation. Despite the extensive collection of total cases, it is crucial to acknowledge the prolonged outbreak with up to five waves, which may pose challenges in establishing statistical significance across various regions during specific waves.

## 5. Conclusions

This study investigated SARS-CoV-2 dynamics in human and feline populations during five outbreak waves in Thailand. The fourth wave had the highest number of human cases, primarily in Bangkok–Pathum Thani. Cat sera were screened using ELISA, reflecting human case trends. No significant effects of epidemic waves or provinces on ELISA-positive cats were found. cPass™ confirmed a 3.99% cat seropositivity rate. Correlations between positive cat sera and human cases varied in significance. The Omicron variant may explain decreased cat cases post-fourth wave. Despite challenges in statistical significance during the prolonged outbreak, the study highlights cats as potential sentinels and emphasizes comprehensive testing methods for accurate assessments in outbreaks.

## Figures and Tables

**Figure 1 animals-14-00761-f001:**
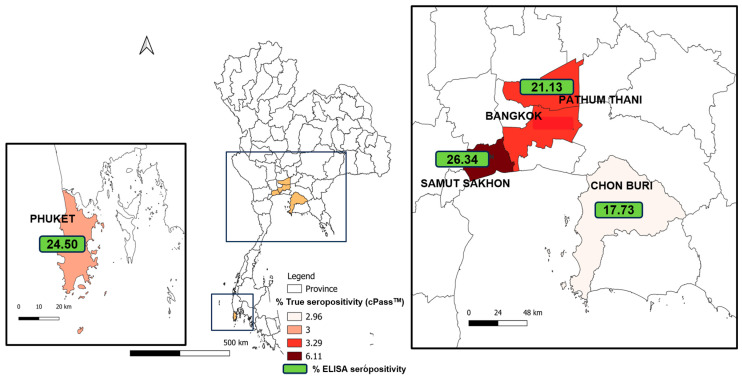
Map of Thailand showing the studied provinces and highlighting the distribution of SARS-CoV-2-positive cat sera detected by cPass™ (% true seropositivity) and the percentages of SARS-CoV-2-positive cat sera detected by indirect ELISA (green).

**Figure 2 animals-14-00761-f002:**
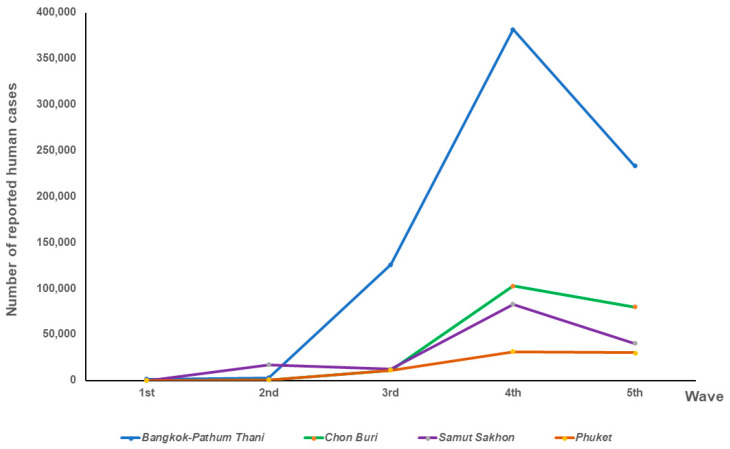
Number of reported human cases in relation to the epidemic’s waves in 5 provinces.

**Figure 3 animals-14-00761-f003:**
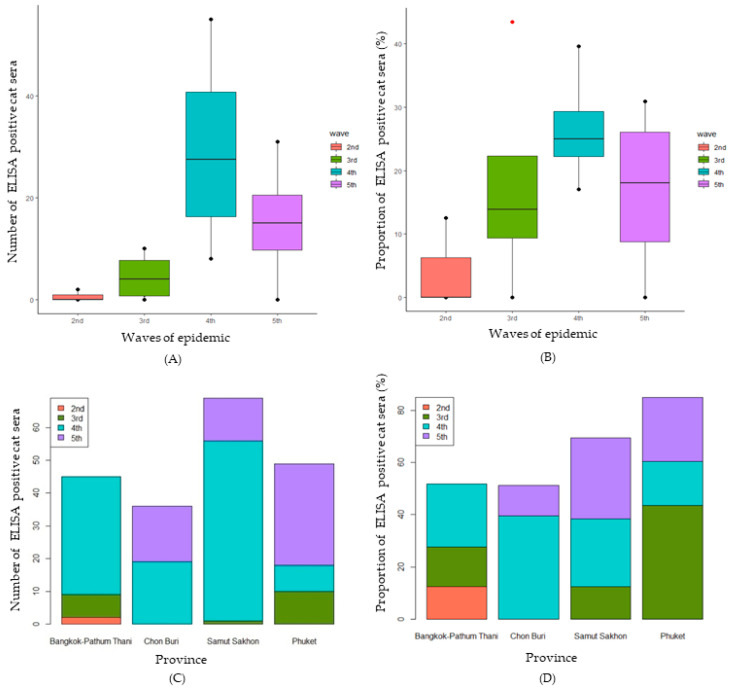
Numbers of ELISA-positive cat serum samples in relation to waves of the epidemic (**A**) and province (**C**) and proportions of ELISA-positive cat sera separated in relation to waves of the epidemic (**B**) and province (**D**).

**Figure 4 animals-14-00761-f004:**
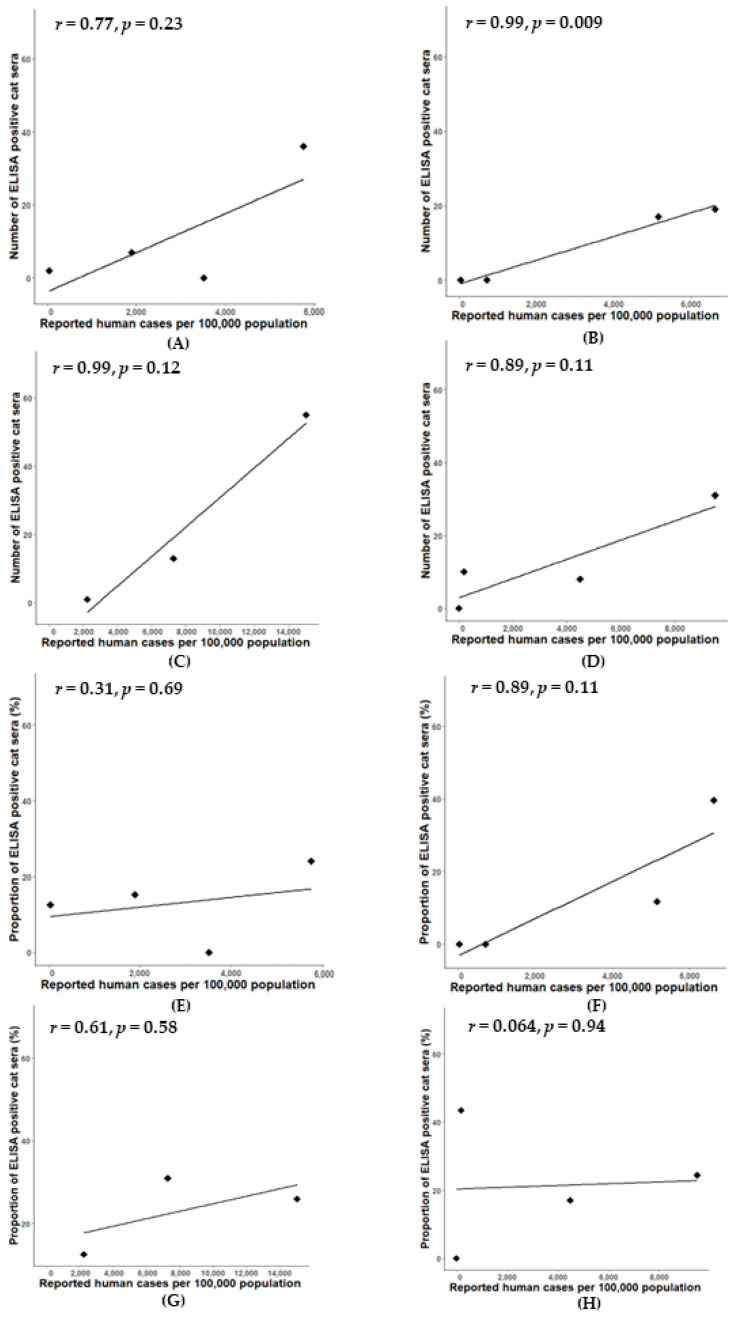
Correlation between number of reported human cases and number of indirect ELISA-positive cat serum samples in four locations: Bangkok–Pathum Thani; r = 0.77, *p* = 0.23 (**A**), Chon Buri; r = 0.99, *p* = 0.009 (**B**), Samut Sakhon; r = 0.98, *p* = 0.12 (**C**) and Phuket; r = 0.89, *p* = 0.11 (**D**) and correlation between number of reported human cases and the proportion of indirect ELISA-positive cat sera in four locations: Bangkok–Pathum Thani; r = 0.31, *p* = 0.69 (**E**), Chon Buri; r = 0.89, *p* = 0.11 (**F**), Samut Sakhon; r = 0.61, *p* = 0.58 (**G**) and Phuket; r = 0.064, *p* = 0.94 (**H**).

**Figure 5 animals-14-00761-f005:**
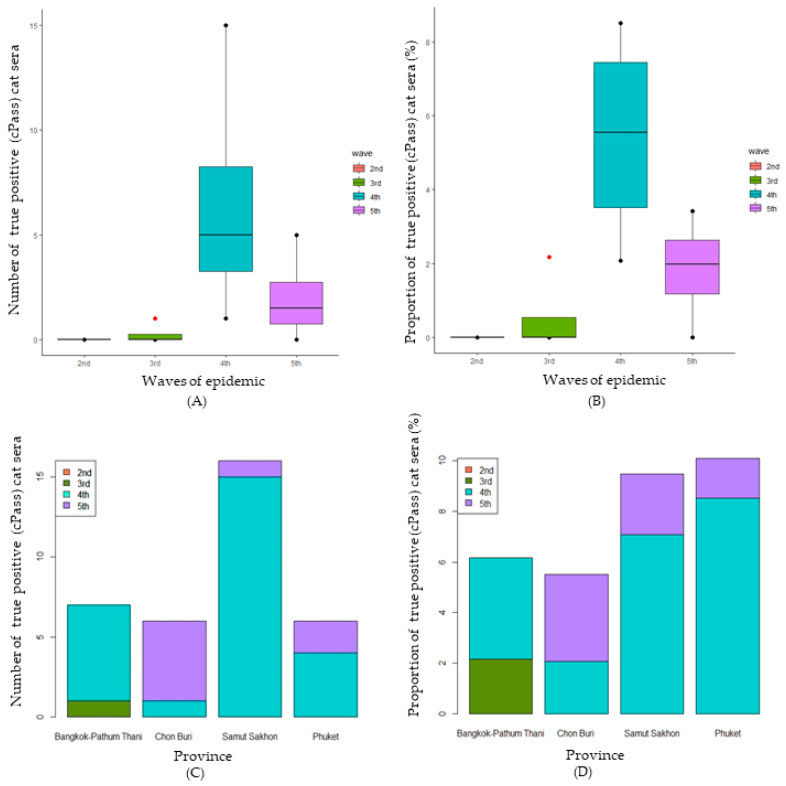
Number of true positive cat serum samples (cPass™) in relation to waves of the epidemic (**A**) and province (**C**) and the percentages of true positive cat serum samples (cPass™) separated in relation to waves of the epidemic (**B**) and province (**D**).

**Figure 6 animals-14-00761-f006:**
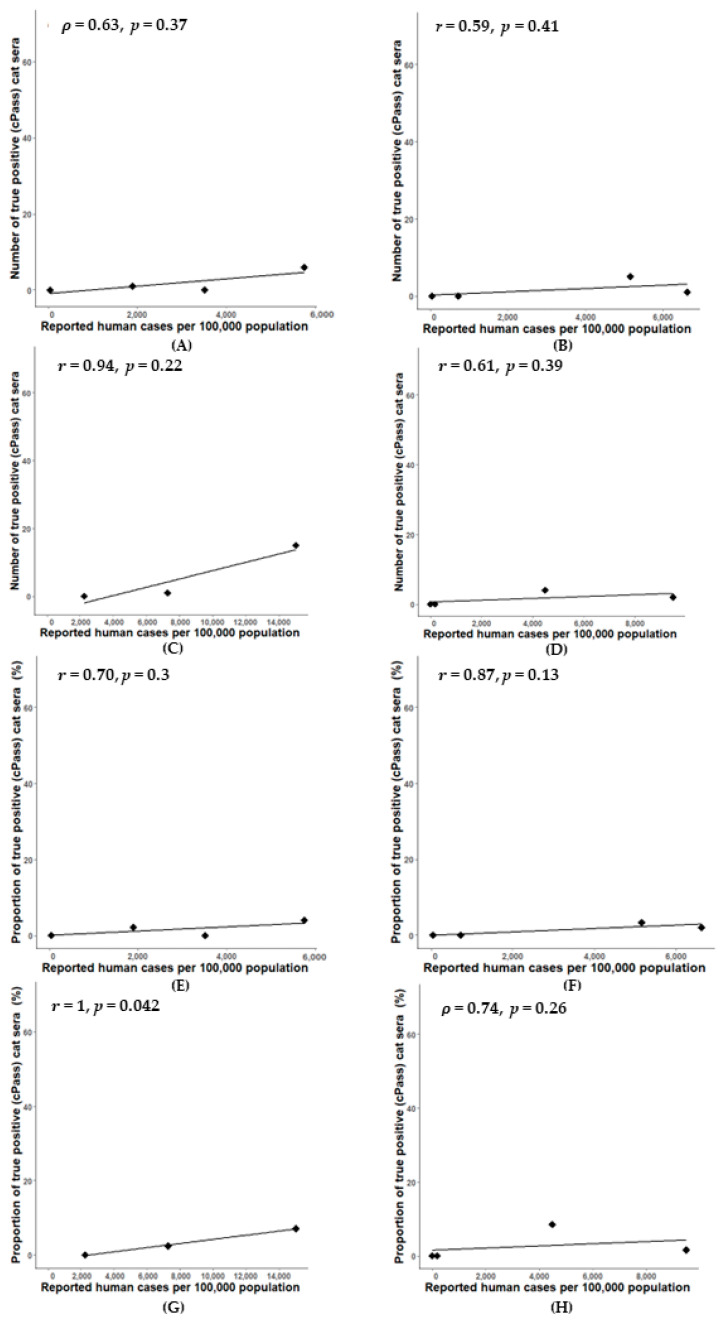
Statistical analyses of the prevalence of reported human cases and number of true positive cat sera (cPass™) in four locations: Bangkok–Pathum Thani; ρ = 0.63, *p* = 0.37 (**A**), Chon Buri; r = 0.59, *p* = 0.41 (**B**), Samut Sakhon; r = 0.94, *p* = 0.22 (**C**), and Phuket; r = 0.61, *p* = 0.39 (**D**), and of the prevalence of reported human cases and percentages of true positive cat sera (cPass™) in four locations: Bangkok–Pathum Thani; r = 0.70, *p* = 0.30 (**E**), Chon Buri; r = 0.87, *p* = 0.13 (**F**), Samut Sakhon; r = 1, *p* = 0.042 (**G**), and Phuket; ρ =0.74, *p* = 0.26 (**H**).

**Table 1 animals-14-00761-t001:** Number of collected cat serum samples and percentage of indirect ELISA-positive cat sera in relation to the reported human cases with severe acute respiratory syndrome coronavirus 2 (SARS-CoV-2) during the 1st to 5th waves of the epidemic in 5 provinces of Thailand from December 2020 to March 2022.

Location	Epidemic Wave	No. of Cat Serum Samples	No. of Positive ELISA	Percentage of Positive Samples by ELISA	No. of Positive cPass Tests	Percentage of Positive Samples by cPass Test	No. of Reported Human Cases	Human Cases/ 100,000 Population
Bangkok–Pathum Thani	1st	0	0	0	0	0	1336	19.99
	2nd	16	2	12.50	0	0	2943	44.38
	3rd	46	7	15.22	1	2.17	125,788	1896.80
	4th	150	36	24.00	6	4.00	381,766	5756.77
	5th	1	0	0	0	0	233,372	3519.09
	Total	213	45	21.13	7	3.29	745,205	11,237.18
Chon Buri	1st	0	0	0	0	0	71	45.92
	2nd	5	0	0	0	0	662	42.61
	3rd	4	0	0	0	0	11,254	724.40
	4th	48	19	39.58	1	2.08	102,794	6616.67
	5th	146	17	11.64	5	3.42	80,052	5152.81
	Total	203	36	17.73	6	2.96	194,833	12,541.06
Samut Sakhon	1st	0	0	0	0	0	14	2.54
	2nd	0	0	0	0	0	17,109	3100.20
	3rd	8	1	12.50	0	0	12,407	2248.19
	4th	212	55	25.94	15	7.08	82,921	15,025.54
	5th	42	13	30.95	1	2.38	40,194	7283.28
	Total	262	69	26.34	16	6.11	152,645	27,659.74
Phuket	1st	0	0	0	0	0	209	51.72
	2nd	3	0	0	0	0	3	0.74
	3rd	23	10	43.48	0	0	769	189.77
	4th	47	8	17.02	4	8.51	18,164	4482.34
	5th	127	31	24.41	2	1.57	38,414	9479.44
	Total	200	49	24.50	6	3.00	57,559	14,203.86

## Data Availability

The datasets generated and/or analyzed during the current study are available from the corresponding author upon reasonable request. The human case data are available from the Department of Disease Control (DDC), Ministry of Public Health of Thailand (https://covid19.ddc.moph.go.th/, accessed on 10 January 2024).

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
