# Peer review of "Seroprevalence of Anti-SARS-CoV-2 Antibodies in Cats during Five Waves of COVID-19 Epidemic in Thailand and Correlation with Human Outbreaks"

_animals, 2024, doi:10.3390/ani14050761_

Round 1
Reviewer 1 Report
Comments and Suggestions for Authors
The authors reported SARS-CoV2 seroprevalence in cats in Thailand during five COVID-19 waves. The manuscript is overall well written. This reviewer has following suggestions for the authors:
1. L63-65: This is great information. Any references for this?
2. L69-70: "The necessity to segregate cats from COVID-19 patients becomes a pivotal measure to avert and mitigate transmission from humans to felines."Any references?
3. L79-80: "recent SARS-CoV-2 past infection". Delete 'past'.
4. Table 1 should be in the results section.
5. L239: SARS-CoV2.
Comments on the Quality of English Language
Minor editing may be needed.
Reviewer 2 Report
Comments and Suggestions for Authors
The topic of the study is interesting and the results of the study could be of potential importance. However, I have some reservations about materials and methods section and especially to statistical analyses. It is my opinion that this study would be suitable for publication with significant revision of the data analysis.
Introduction
The introduction is well made, but the goal of the study at lines 94-98 should be better explained.
Materials and methods
Line 102: What does the number 136 in the bracket represent (1107-136)? Please explain.
Results
Line 176-178: Figure 1. Please make the maps more clearly visible. Furthemore, replace the phrase “% true seropositive” with “% true seropositivity”. Similarly, replace the phrase “% ELISA positive” with the phrase “% ELISA seropositivity”.
Line115-118: Has the modified SARS-CoV-2 S1 RBD ELISA been used again in another study? Please cite the relevant article.
Line 140-142: Has the “cPass” used again in another study? Please cite the relevant article
Statistical analyses. In total, in data analysis only the cat samples that were positive for SARS-CoV-2 infection cPass surrogate viral neutralization (confirmation test) should be used,as the only reliable. The analysis and the results of cat samples performed by the modified SARS-CoV-2 S1 RBD ELISA can cause confusion to the reader. Furthermore is more useful for interpreting only the correlation between the prevalence of seropositivity in cats and human
Line 154: Rephrase: “to determine the prevalence of seropositivity in cats”
Line 162-164. Table 1. Why isn’t there the prevalence of seropositivity in human? Correlation test between the proportion of positive cat sera (cPass confirmation test) and proportion of positive human cases in four locations should be performed
Line 181: Typo in SARS-CoV-2
Line 184-186: Add this sentence to Materials and Methods section
Discussion
Lines 325-327: Add reference
Comments on the Quality of English LanguageMinor editing of English language required
Reviewer 3 Report
Comments and Suggestions for Authors
It is very important to understand the correlation between the distribution of SARS-CoV-2 antibodies in cats during the pandemic and the number of confirmed cases in humans is crucial. It can provide valuable insights for the management of zoonotic diseases that affect both humans and animals.
Overall concerns and questions:
1. Certain terms employed in this manuscript are not entirely appropriate; for instance, "Antropozoonosis" should be replaced with "zoonosis." Additionally, in line 26, the phrase "a 22.67% seropositivity rate" might be better expressed as "the seropositivity rate is 22.67%" or "seropositive rate of 22.67%" for clarity and precision.
2. From lines 74 to 77, the authors introduced a potential alternative approach to monitor COVID-19 by using the cat population as sentinels and referenced works 11 and 12. However, these references primarily address the endemic transmission of COVID-19 rather than specific cat population monitoring programs. To improve clarity, it is suggested that references 11 and 12 be cited immediately following the sentence "…disease reporting may decrease."
3. Would it be really more practical to monitor SARS-CoV-2 in the cat population instead of implementing a routine human population monitoring program?
4. The antibodies generated in SARS-CoV-2-infected cats can include both neutralizing and non-neutralizing antibodies. What is the rationale for employing the cPass surrogate viral neutralization test for "confirmation"? Don’t the non-neutralizing antibodies belong to antibodies against SARS-CoV-2? What is the rational for false positive cases due to the reduction mentioned in line 322?
5. How about the transmission of SARS-CoV-2 between cat to cat?
6. Would the introduction of SARS-CoV-2 into cat populations increase the possibility of recombination between SARS-CoV-2 and existing feline coronaviruses, leading to the emergence of novel coronaviruses?
M&M Section:
1. What is sources of cats, household or shelter? What is the interaction between the cats and humans?
2. In line 102, what is the meaning of “1107-136”?
3. In line 122, which temperature is really used, 37oC or room temperature?
4. In line 124, what is the temperature for “After 30 minutes of incubation”?
5. In line 125, the incubation temperature after adding TMB is really 37oC?
6. In line 128 and 148, what is the composition of stop solution?
7. In line 129, what is the meaning of “18 categories”?
8. In line 151, how to determine the threshold for neutralization is 20%?
Discussion section:
1. In line 325, what is the rationale or supporting references for the theory suggesting that false positive antibody responses were potentially caused by the interference of co-infection with other feline viruses?
Round 2
Reviewer 2 Report
Comments and Suggestions for Authors
Accept in present form